# Pectins Rich in RG-I Extracted from Watermelon Peel: Physicochemical, Structural, Emulsifying, and Antioxidant Properties

**DOI:** 10.3390/foods13152338

**Published:** 2024-07-25

**Authors:** Xiaojun Ma, Xinxin Cheng, Yuyi Du, Peiyao Tang, Liangxiao Chen, Wei Chen, Zhenjia Zheng

**Affiliations:** 1College of Food Science and Engineering, Shandong Agricultural University, Taian 271018, China; maxj003@163.com (X.M.); dyyyyds1999@163.com (Y.D.); purelstpy@163.com (P.T.); 18660830186@163.com (L.C.); zhengzhenjia@sdau.edu.cn (Z.Z.); 2College of Agronomy, Shandong Agricultural University, Taian 271018, China; chengxx0120@163.com

**Keywords:** RG-I structural domain, pectic polysaccharides, emulsifying properties, watermelon peel, antioxidant

## Abstract

RG-I pectin has excellent health benefits, but its raw materials are relatively scarce, and its complex structure often breaks down its side-chain structure during the extraction process. In this study, the physicochemical and antioxidant properties of a branched-chain-rich pectin gained from watermelon peel were demonstrated, and the structure–function relationships of RG-I-enriched pectin and emulsification properties were investigated. Fourier transform infrared spectroscopy, high-performance anion exchange chromatography, high-performance gel permeation chromatography, nuclear magnetic resonance spectroscopy, and methylation analyses reveal it as acetylated, low-methoxylated pectin, rich in RG-I side chains (MW: 1991 kDa, RG-I = 66.17%, methylation degree: 41.45%, (Ara + Gal)/Rha: 20.59%). RPWP outperforms commercial citrus pectin in emulsification and stability, significantly preventing lipid oxidation in emulsions. It also exhibits free radical scavenging abilities, contributing to its effectiveness in preventing lipid oxidation. Emulsions made with RPWP show higher viscosity and form a weak gel network (G′ > G″), enhancing stability by preventing phase separation. These findings position watermelon peel as a good source of RG-I pectin and deepen our understanding of RPWP behavior in emulsion systems, which may be useful in the food and pharmaceutical fields.

## 1. Introduction

Pectin is a complex polysaccharide widely distributed in plant cell walls and inner cell layers. Based on its major structural domains, pectin can be categorized into highly branched rhamnogalacturonan-I (RG-I), rhamnogalacturonan-II (RG-II), and homogalacturonan (HG), the latter of which is primarily linear in structure [1]. Recently, RG-I pectin has gained increasing attention due to the positive contributions of its side chains to gelation, emulsification, properties, and health benefits—advantages that HG pectin lacks—making it a suitable natural alternative for food and pharmaceutical applications [2,3]. In addition, the potential of pectin as a prebiotic has been widely proven. It helps lower blood pressure and improve intestinal flora. Using pectin as a natural and efficient emulsifier in food will improve food quality and give it healthy properties [3]. Currently, much research focuses on identifying new raw materials or extraction methods to obtain RG-I pectin more easily, especially from low-cost waste byproducts [4]. Traditional pectin extraction processes often involve high temperatures and strong acids/bases, which can hydrolyze glycosidic bonds and severely disrupt side-chain structures [5]. Non-thermal extraction techniques (e.g., microwaves, ultra-high pressure, and enzymes) have been explored, but they present challenges such as high energy consumption, low extraction efficiency, and high maintenance costs [6].

Watermelon (*Citrullus lanatus*) is a significant cash crop in temperate regions, with substantial annual output. The rind accounts for about one-quarter of the watermelon’s weight, yet large amounts are discarded as waste, consuming resources for disposal and causing environmental issues. Studies have shown that watermelon rind is rich in pectin polysaccharides and is a good source of antioxidant phenolics [7,8]. Therefore, extracting RG-I pectin with favorable properties from watermelon rind can provide economic benefits and make it a valuable source for pectin extraction. Although much research has focused on extracting pectin from watermelon rind, few studies have specifically examined the extraction of RG-I pectin from watermelon peel and its structure–function relationships. Thus, obtaining RG-I pectin from watermelon peel and understanding the interaction between its structure and emulsifying properties is crucial for the high-value utilization of watermelon peel and the broader application of RG-I pectin.

In this study, RG-I-enriched pectic polysaccharides were extracted from watermelon peel (RPWP) using a mild hot acid method. Their emulsification and the antioxidant properties of RPWP and commercially available citrus pectin (CP) were compared. The monosaccharide composition and molecular weight of the pectic polysaccharides were determined using high-performance anion exchange chromatography (HPAEC) and gel permeation chromatography (HPGPC), and the polysaccharide structures were characterized by Fourier transform infrared (FTIR) spectroscopy, 1D and 2D NMR analysis, and glycosidic bond analysis to confirm the RG-I structural domain in RPWP. The microscopic characteristics, three-phase contact angle, and antioxidant capacity of RPWP were also analyzed and compared with CP to explore the connection between the RPWP structure and its gel and emulsification properties. This study demonstrates the feasibility of extracting RG-I-rich pectin from watermelon rind and compares its emulsifying properties with those of commercially available citrus pectin. By analyzing the structure of RPWP, we elucidate the relationship between the RG-I domain and the emulsifying and antioxidant properties of pectin. The findings highlight watermelon rind as a promising and economical source for RG-I pectin extraction. This research provides valuable insights into the structure–function relationships of RG-I-rich pectin, underscoring the potential of RPWP as a natural ingredient for applications in the food and pharmaceutical industries.

## 2. Materials and Methods

### 2.1. Materials

Watermelons (same size, maturity, and ripeness, about 4.5 kg) were supplied by the College of Food Science and Engineering (Shandong Agricultural University, Taian, China) and harvested during July and August. The peel (without pulp) was sliced into a certain size and then dried by a freeze-dryer (SCIENTZ-12, Ningbo Scientz Biotechnology Co., Ltd., Ningbo, China), and finally, watermelon peel powder was obtained by a high-speed grinder and 200-mesh sieve.

Citrus pectin was purchased from Andre group (Yantai, China). Molecular weight standards were purchased from Shanghai yuanye Bio-Technology Co. (Shanghai, China), Ltd. and Sigma-Aldrich Corporation (St. Louis, MO, USA). Monosaccharide samples (standard) were purchased from Bop Rui Saccharide Biotech Co. Ltd. (Yangzhou, China). Folin–Ciocalteu’s reagent (F8060) was purchased from Solarbio (Beijing, China), and calf serum (C1401) was purchased from Pingrui Biotechnology (Beijing, China). All other reagents used in this study were analytical grade.

### 2.2. Pectin Extraction and Purification

The method of extraction and purification from the study of Zhang [9] was performed with some modifications. Watermelon peel powder was mixed with deionized water at a ratio of 1:20 (*m*/*v*), and the pH of the solution was adjusted to 2 using citric acid (1 M). The mixture was stirred at 700 rpm for 4 h in a water bath 60 °C and then centrifuged at 5000× *g* for 15 min, taking the supernatant as the next step. Triple the volume of anhydrous ethanol was added to the centrifuged supernatant, which was then precipitated for 12 h (24 °C), changing water every 6 h. The solution was then centrifuged at 4500× *g* for 15 min, and the precipitated pellets were redissolved in a certain volume of water. The solution was dialyzed in a 3.5 kDa dialysis bag for 48 h (24 °C). Finally, the purified pectin was obtained by freeze-drying.

### 2.3. Physicochemical Property of Pectin

The protein content was determined using the Bradford method [10], and the total polyphenol content of pectin was determined using the Folin–Ciocalteu reagent method [11]. The pectin GalA content was measured using the m-hydroxydiphenyl determination method and verified by the result of monosaccharides. A titration method was used to measure the degree of esterification (DM%) and the methoxyl content (MeO%) of pectin [12].

### 2.4. Structural Characterization of Pectin

#### 2.4.1. Fourier Transform Infrared Spectroscopy

The FTIR spectra of pectin were collected using the FTIR spectra Nicolet iS5 system (Thermo Fisher Scientific, Waltham, MA, USA) and analyzed as described by Zhang et al. [9]. The scan was acquired over a range of 4000–400 cm^−1^ and was performed 64 times with a resolution of 4 cm^−1^ against the background. As described by Jittra et al. [13], the DM was determined by the average ratio of the peak area of 1740 cm^−1^ (COO-R) to the curve areas of 1740 cm^−1^ to 1630 cm^−1^ (corresponding to the number of total carboxylic groups).

#### 2.4.2. Monosaccharides and Molecular Weight Distribution Analysis

The monosaccharide composition of pectic polysaccharides was analyzed using HPAEC as described by Méndez et al. [14]. A Dionex ICS-5000 (Thermo Scientific, Waltham, MA, USA) equipped with a CarboPac™ PA-20 analytical column (3 × 150 mm) and a pulsed amperometric detector were used for the analysis. Pectin was hydrolyzed using trifluoroacetic acid (TFA) for 3 h at 120 °C. The sample volume was 5 μL, the column temperature was 30 °C, and the flow rate was 0.3 mL/min. The content of each monosaccharide was expressed as mol% of the total content, and the approximate ratios of HG and RG-I were calculated using the contents of GalA, Ara, Gal, and Rha [3].
$$\mathrm{HG}\;\left(\%\right)=\mathrm{GalA}\%\;-\;\mathrm{Rha}\%$$
$$\mathrm{RG}-I\;\left(\%\right)=2\mathrm{Rha}\%+\mathrm{Ara}\%+\mathrm{Gal}\%$$

The molecular weight (MW) of pectin was detected as described by Li et al. [15]. HPGPC (LC-10A, Shimadzu, Kyoto, Japan) was performed with a BRT105-104-102 column (8 × 300 mm) and a refractive index detector (RI-502, Shimadzu, Kyoto, Japan) to measure the MW of pectin. The mobile phase comprised 0.05 M NaCl solution, and the pectin samples were dissolved in the mobile phase (5 mg/mL). The sample volume was 20 μL, the column temperature was 40 °C, and the flow rate was 0.6 mL/min.

#### 2.4.3. NMR Spectroscopy Analysis

Pectin structure was analyzed using 1D (^1^H and ^13^C) and 2D (COSY, HSQC, TOCSY, and NOESY) NMR spectroscopy. The analysis was performed using a Bruker 600 MHz AVANCE III NMR spectrometer (Bruker, Rheinstetten, Germany) equipped with a 5 mm CPP-TCI probe (600 MHz) (Bruker, Billerica, MA, USA). Pectin (50 mg) was dissolved in D2O (0.5 mL), and the ^1^H spectrum was obtained with 320 scans. Approximately 100 mg of the sample was packed into a 4 mm rotor and compacted before being placed in a solid-state NMR probe. The experiment was conducted at a spinning rate of 10 kHz using the Cross-Polarization Magic Angle Spinning (CPMAS) pulse sequence for the ^13^C NMR analysis. The ^13^C resonance frequency was set at 150.91 MHz, with 2000 acquisitions. A two-dimensional spectrum was recorded using standard Bruker procedures.

#### 2.4.4. Glycosidic Linkage Analysis

Glycosidic linkages in pectin were determined using gas chromatography-mass spectrometry (GC-MS) analysis [16]. Briefly, 5 mg of pectin was dissolved with anhydrous dimethyl sulfoxide (DMSO) and methylated with CH_3_I and hydrolyzed using TFA (2M, 1 mL) for 90 min at 121 °C. The hydrolysates were reduced using 60 mg of borohydride sodium with 2 mL double distilled water for 8 h and acetylated using 1 mL acetic anhydride for 1 h at 100 °C. HPAEC was used to determine whether the uronic acids were completely reduced to their equivalent neutral sugars. Additionally, the partially methylated alditol acetates (PMAAs) derived from the reduced samples were analyzed using a GC-MS system (6890-5973, Agilent, Santa Clara, CA, USA) equipped with an RXI-5 SIL MS column (30 m × 0.25 mm × 0.25 µm) to further investigate the structural characteristics of the methylated sugars.

### 2.5. Three-Phase Contact Angle

The three-phase contact angle was determined using an optical contact angle meter (Powereach Instruments Ltd., Shanghai, China). The pectin was prepared as pressed slices, completely submerged in corn oil (Xiwang Group Limited, Binzhou, China), and analyzed by filming with a high-speed camera after 2 μL of deionized water was dropped onto the slice surface at a constant rate.

### 2.6. Preparation of Emulsion

Preliminary experiments were conducted to determine the optimal concentration of the emulsions. The RPWP and CP were dissolved in deionized water at different concentrations (0.5%, 1.0%, 1.5%, 2.0%, and 4% *w*/*v*) and stirred for 4 h at room temperature to obtain hydrated pectin. The pectin solutions were mixed with equal volumes of corn germ oil using a high-speed homogenizer (IKA T18, IKA-Werke GmbH & Co. KG, Staufen, Germany) at 15,000 rpm for 5 min to prepare the emulsions [17].

#### 2.6.1. Emulsifying Properties

The emulsifying capacity (EC) and emulsion stability (ES) were determined as described by Lu et al. and Zhan et al. [9,18] The EC, ES, and CI were evaluated using the following equations:$$\mathrm{EC}(\%)\;=\frac{Ve}{Vm}\text{×}100\%$$
$$\mathrm{ES}\left(\%\right)\;=\frac{Vh}{Ve}\text{×}100\%$$
where V_e_ is the volume of the emulsion after the first centrifugation, V_m_ is the total volume of the emulsion, V_h_ is the emulsion volume remaining after high-temperature storage.

#### 2.6.2. Emulsion Droplet Size Distribution and Optical Microscopy Observation

The emulsion droplet size distribution was analyzed by laser particle size analyzer (Zetasizer Nano S90, Malvern Instruments Ltd., Worcestershire, UK). Thirty μL of the emulsion was placed on a microscope glass slide and covered with a coverslip and observed by IX73 fluorescence inversion microscope system (Olympus, Tokyo, Japan) at room temperature to obtain the microstructures of emulsion.

### 2.7. Rheological Properties

RPWP and CP were dissolved in distilled water, stirred for 8 h, then placed at 4 °C for 24 h to obtain pectin solutions (2% and 4%, *m*/*v*). The viscosity, storage modulus (G′), and loss modulus (G″) of emulsions and pectin solution were determined by a controlled stress rheometer (AR2000ex, TA Instruments, New Castle, DE, USA) equipped with a 20 mm parallel plate configuration at 25 °C. The controlled shear rate mode (CR) was set such that the shear rates ranged from 0.1 to 100 s^−1^. The controlled deformation mode (CD) was set to a frequency range of 0.02–10 Hz. All data were obtained by averaging the results of three independent experiments.

### 2.8. Antioxidant Capacities

Vitamin C (Solarbio, Beijing, China) was taken as a control. The scavenging activities on DPPH and hydroxyl radicals of 5, 10, 15, and 20 mg/mL pectin solution were measured by the assay kit (BC475, Solarbio, Beijing, China) at 25 °C. The malondialdehyde (MDA) Assay kit (BC0025 and BC1325, Solarbio, Beijing, China) was used to measure the MDA content in the corn germ oil, pectin solution, and emulsions in 30 days.

### 2.9. Statistical Analysis

All data in this study are the mean value and standard deviations of at least three measurements. Statistical analysis was carried out by SPSS software 25 (SPSS Inc., Chicago, IL, USA), and the results were analyzed by one-way ANOVA and Duncan’s multiple-range test. The difference was statistically significant at *p* < 0.05.

## 3. Results and Discussions

### 3.1. Chemical Composition and Molecular Weight

RG-I-enriched pectic polysaccharides from watermelon peels were obtained through water extraction, precipitation, and freeze-drying. Their chemical composition and molecular weight were then determined. The degree of methylation (DM) of RPWP and CP was analyzed using a titrimetric method and confirmed via FTIR, revealing DM values of 41.45% for RPWP and 58.48% for CP, indicating that RPWP is a low-methoxy pectin. The protein and total polyphenol contents of RPWP were determined through chemical titration, measuring 6.1% and 2.4%, respectively. Protein and phenolic content are often considered key factors in emulsion stability [6]. The combination of proteins and polysaccharides can enhance emulsification properties, while the presence of polyphenolic substances can provide additional antioxidant activity and inhibit lipid oxidation in emulsions [17]. Thus, the presence of small amounts of proteins and polyphenols in RPWP is beneficial for improving emulsification performance and emulsion stability.

The monosaccharides in pectin are presented in Table 1 and Appendix A. Rha, Ara, Gal, Glc, and GalA (2.9%, 30.7%, 29.6%, 3.4%, and 27.9%) were the main monosaccharides in the RPWP, consistent with the previous literature [19]. The amount of Xyl in RPWP was small and may have been derived from xylogalacturonan. This result indicates few xyloglucan structures in RPWP [20]. The ratio of (Ara + Gal)/Rha reflects the size of the arabinose and galactose side chains, the ratio of Rha/GalA reflects the main chain of pectin, and GalA/(Rha + Ara + Gal + Xyl) indicates the linearity of pectin. Notably, RPWP had a lower linearity (0.43%) and greater degree of branching (20.59%) than that of CP (1.2% and 3.65%). The RG-I content of RPWP was 66.17%, which was higher than that of CP (35.00%). Meanwhile, the HG molar ratios of RPWP and CP were 24.97% and 44.27%, respectively, indicating that more abundant side-chain structures existed in RPWP [9]. These results indicate that RPWP is composed of abundant arabinose and galactose side chains in the RG-I domain, which may be attributed to the gentle extraction conditions causing less destruction of the pectin structure in watermelon peel [21].

The Mw, Mn, and Mw/Mn ratios are listed in Table 1. Unexpectedly, the molecular weight of RPWP (1991 kDa) is almost 3.5 times greater than that of CP and significantly higher than the watermelon rind pectin obtained by common hydrochloric acid extraction (106 kDa) [22], indicating that the mild thermal acid extraction process better preserved the structure of pectin. Citric acid treatment is preferred to protect and polymerize pectin with minimal degradation [9]. The small amount of Glc detected using HPAEC (Table 1) may be attributed to under-degraded cellulose, and the galactose or arabinose units in the RG-I side chain may be covalently attached to the plant cellulose, suggesting the possible formation of a pectin-cellulose macropolymer. Furthermore, proteins and polyphenols can attach to pectin and form macromolecular complexes during the extraction process [23]. A larger molecular weight has been reported to impart higher viscosity to pectin, which is beneficial for its rheological properties and emulsion stability [9].

### 3.2. Glycosidic Linkage Analysis

The results of methylation analysis, as presented in Appendix A, reveal that RPWP is characterized by a rich diversity of polysaccharides, encompassing at least 5 types and 12 distinct linkage patterns. Notably, the linkage patterns →4)-Galp-(1→(34.2%) and terminal Galp-(1→(9.5%) are likely derived from native galactose and reduced galacturonic acid. The potential underestimation of these values, due to incomplete reduction, suggests an even greater prevalence of these linkages. Significantly, these patterns, along with →2,4)-Rhap-(1→(2.7%) and →3,4)-Rhap-(1→(2.8%), constitute the backbone of RPWP, affirming its RG-I domain composition [24]. The predominance of the →4)-Galp-(1→ and →2,4)-Rhap-(1→ linkages further corroborates the extensive RG-I region in RPWP.

The degree of branching (DB), calculated as (N_T_ + N_B_)/(N_B_ + N_T_ + N_L_) [25], where N_T_, N_B_, and N_L_ represent the molar ratios of terminal, branching, and linear residues, respectively, is 0.774. This high DB value underscores the complexity of the side-chain structure within the RG-I region of RPWP, aligning with the conclusions drawn in Section 3.1. The arabinose linkage patterns, including Araf-(1→, →5)-Araf-(1→, →3)-Galp-(1→, →6)-Galp-(1→, and →3,6)-Galp-(1→, suggest the presence of AG-II and specific arabinan structures as side chains in the RG-I region [26]. This comprehensive analysis highlights RPWP as a pectin rich in RG-I domains, featuring an intricate network of side chains.

### 3.3. FTIR and NMR Analysis

The FTIR results, as depicted in Figure 1A, reveal characteristic polysaccharide absorption bands for RPWP and CP around 3400 cm^−1^, 2950 cm^−1^, 1700 cm^−1^, and 1100 cm^−1^. The absorption bands at 1650 cm^−1^ and 1434 cm^−1^, attributed to the C=O stretching vibrations, indicate the presence of uronic acids in the pectin, aligning with the monosaccharide composition analysis [27].

The 1D and 2D NMR spectra, presented in Figure 1 and Appendix A, show typical polysaccharide signal patterns in the range of 3.0–5.5 ppm for ^1^H and 60–110 ppm for ^13^C. Due to the high molecular weight and low sensitivity, the assignment of ^1^H/^13^C signals was incomplete. The partial chemical shifts of residues in the NMR spectra for RPWP were assigned regarding the literature and the results of glycosidic linkage analysis, as listed in Appendix A [24,28,29,30].

In the ^1^H spectrum, signals ranging from 4.9 to 5.4 ppm are likely due to the alpha-glycosidic form, while signals from 4 to 4.9 ppm suggest the presence of beta-anomeric compounds. The ^13^C spectrum, with signals in the range of 90–110 ppm, corroborates the coexistence of both alpha and beta configurations in RPWP, consistent with the 1H spectrum findings [3]. The signals at 4.93 ppm, 3.52 ppm, 3.84 ppm, 4.23 ppm, and 4.69 ppm was attribute to the H1, H2 H3, H4, and H5 of GalA [29], indicating the galacturonan backbone in RPWP. The HSQC spectrum shows signals at 53.72/3.75 ppm, attributed to CH_3_O groups in 1,4-α-D-GalpA (OMe), and the COSY spectrum reveals adjacent proton couplings on GalpA(OMe) residues (H1/H2, H2/H3, and H4/H5). Furthermore, the NOESY (Appendix A) spectrum displays inter-/intra-residual coupling signals of H1/H4 of →4)-α-GalpA-(1→ at 4.93/4.23 ppm, indicating the presence of the →4)-α-GalpA-(1→4)-α-GalpA-(1→ fragment [24].

For the analysis of →4)-α-GalpA-(1→, rhamnopyranosyl residues were assigned using 1D and 2D NMR spectra. In the 1H spectrum, signals around 1.15 ppm correspond to the proton resonances of methyl groups in rhamnose, with a corresponding resonance at 16.78 ppm in the ^13^C spectrum for RPWP. In 1,2,4-linked Rhap residues, the methyl groups in rhamnose were assigned to C-6, confirmed by the cross peak at 16.78/1.11 ppm in the HSQC spectrum [29]. The HSQC signals at 77.57/4.10 ppm and 83.41/3.88 ppm were attributed to C2/H2 and C4/H4 of →2,4)-α-Rhap-(1→, with C3 signals at 74.93 ppm confirming the substitution positions in α-Rhap residues. Additionally, the NOESY (Appendix A) spectrum cross peak between H1 of 1,4-α-D-GalpA and H2 of α-L-Rhap at 4.93/4.10 ppm, and between H4 of 1,4-α-D-GalpA and H1 of α-L-Rhap at 4.23/4.96 ppm, indicates the presence of the RG-I region [29]. The presence of Araf-(1→, →5)-Araf-(1→, and →3,5)-Araf-(1→ is assigned to the arabinan with a 1,5-linked α-Araf backbone and branch points at O-3 position (Appendix A). The COSY spectrum cross peaks (H1/H2, H2/H3, H3/H4, H4/H5) indicate the correlations of these substituted α-L-Araf residues. Other small signals not assigned are derived from xylose and glucose.

The NMR analysis results indicate that RPWP is a pectin polysaccharide primarily composed of 1,4-α-D-GalA as the main uronic acid, with 1-, 1,5-α-L-Ara, and 1,2,4-Rha as the main residues featuring HG and RG-I structures. This finding is consistent with the results of methylation analysis. In this structure, rhamnose predominantly exists in 1,2,4-linkage and 1,3,4-linkage forms. The abundant side-chain structures are mainly composed of arabinose and galactan, connected to the C3 or C4 of rhamnose in the RG-I region.

### 3.4. Wettability and Scanning Electron Microscopy (SEM) Analysis of RPWP and CP

The differences between the microscopic morphologies of RPWP and CP were further demonstrated using SEM (Figure 2). RPWP was smooth with a small amount of lamellar structures; this compact structure provides a better deformation resistance to pectin, enhancing the stability of the pectin network. According to the previous study, the smooth layer structure brings better gel strength due to the more load-bearing space of pectin [31]. These findings may be attributed to the re-assembly of pectin during the extraction period, which makes it easier for pectin to cross-link in solution and form a more compact structure. CP in solution constitutes a loose and disordered morphological structure owing to its low MW and branching degree (compared to RPWP) [32]. These results support the results of the structural analysis.

The results of the three-phase contact angle for RPWP and CP are shown in Figure 3D, E. The three-phase contact angle of RPWP (96.77°) was closer to 90° than that of CP (107.46°); this suggests that RPWP has good interfacial amphiphilicity and may exhibit better stability in emulsion systems. In general, hydrophobic groups such as acetyl groups impart hydrophobicity to pectin, whereas the three-phase contact angle directly corresponds to the oil–water interface properties of pectin [33].

### 3.5. Analysis of Rheological Properties

As shown in Figure 3C, the viscosities of CP and RPWP increased with increasing solution concentration and decreased with an increase in the shear rate. The increased shear force led to reduced entanglement between pectin sugar chains, shear thinning, and reduced viscosity; further, their flow behavior indices were both less than 1, suggesting the shear-thinning behaviors of non-Newtonian fluids. According to a previous report, a decrease in the inter-entanglement of pectin chains and an increase in shear rate are closely related to shear thinning fluidity, finally leading to a decrease in apparent viscosity [9]. At the shear rate ranging from 0.1 to 100 s^−1^, the RPWP showed a higher viscosity than that of CP at the same solution concentration. This may be attributed to the abundance of RG-I side chains in RPWP and its higher molecular weight. The decrease rate of viscosity in RPWP was lower than that in CP throughout the shear rate range, indicating the better anti-shear ability of RPWP compared to that of CP; notably, according to Zhang et al., high viscosity contributes to emulsion stability [34].

As shown in Figure 3B,C, the 2% CP solution exhibited fluidic properties at vibration frequencies from 0.1 to 100 Hz with G′ less than G″. The intersection of the G′ curve with the G″ curve occurred at 0.398 Hz and 1.58 Hz vibrational frequency in the 2% RPWP solution and 4% CP solution, respectively, indicating disruption of the gel network in the solution. In contrast, G′ in the 4% RPWP solution was constantly larger than G″ and exhibited a solid-like nature, indicating the transformation of RPWP solution from a liquid- to solid-like state with increasing concentration, demonstrating the potential of RPWP as a gelling agent, which is different from CP. In highly concentrated RPWP pectin solutions, owing to the rich RG-I structural domain of RPWP, intermolecular chain entanglement is formed easily, thus resulting in a three-dimensional network structure and imparting the solution with solid-like properties.

### 3.6. Emulsifying Properties of RPWP

#### The Visual Appearance, Microstructure, and Particle Size Distribution of Emulsions

We also prepared emulsions with CP and RPWP as emulsifiers (CPE and RPWPE, respectively) to evaluate the emulsifying potential of RPWP and CP, with results presented in Figure 4 and Figure 5. Figure 4 illustrates that RPWP exhibits higher EC and ES values than CP, with a statistically significant difference observed between the two. Additionally, within the concentration range of 0.5% to 2%, emulsions of the same pectin displayed significant variations in EC and ES values, indicating a concentration-dependent behavior. These findings suggest that RPWP possesses a superior emulsifying potential compared to CP. Obvious phase separation occurred within 1 h for the 0.5% and 1.0% CPE, and severe phase separation occurred for all concentrations of CPE after 45 days of storage. In contrast, RPWPE exhibited excellent emulsification performance; only 0.5% RPWPE underwent slight phase separation in 24 h. However, phase separation did not occur even for 2% of RPWPE after 45 d of storage (Figure 5A). The emulsion stability increased with the addition of pectin, which could be due to the higher viscosity of pectin at high concentrations [35]. Compared to CP, RPWP has an abundant RG-I structural domain, and the higher degree of branching entangles the pectic polysaccharides in RPWPE with each other, forming a highly stable structure that can resist the precipitation of water caused by gravity and exhibit better emulsification ability.

To further investigate the differences in the emulsions, the droplet size was examined using a microscope and photographed, and the results are shown in Figure 4C and Figure 5B. As the concentrations increased, a decrease in particle size and an increase in emulsion particles were observed for CPE and RPWPE, respectively. According to the image, demulsification occurred in 0.5% and 1.0% CPE, whereas the particles in RPWPE retained integrity even on the 45th day. High viscosity can help the emulsion resist gravity, and a small droplet size contributes to the stability of the emulsion [35]; the rich branched structure of RPWP in the RG-I region makes it easier to cross-link in the emulsion to maintain a stable structure. Compared with CP, it can form smaller droplets in the emulsion to resist the demulsification caused by gravity, providing better stability for RPWPE. The droplet distribution of the emulsions at 45 days is shown in Figure 4C, and the results are consistent with the microscopic photographs of the emulsions. In CPE, with an increase in storage time, the particle size of the emulsion gradually increased and demulsification occurred. In contrast, the droplets in RPWPE were uniform and stable with no demulsification at any concentration during the storage cycle; further, the change in droplet size over time was less than that in CPE. The emulsions with higher concentrations had smaller particle size distributions when the same pectin was used, owing to the spatial repulsion between droplets, which increased with concentration. The smaller droplet size of RPWPE compared with that of CPE proves that RPWP can better encapsulate the oil in the emulsion and form a stable emulsion system, owing to its excellent oil–water interfacial properties.

Combining the above data, RPWPE was found to be more stable than CPE, and the emulsification performance of RPWP was better than that of CP. RG-I pectins have complex branched structures and binding sites. Thus, RG-I pectins containing certain proteins and polyphenols are more likely to produce stronger intra- and intermolecular bonds and form complex dense network structures, and the rich branched structure of RPWP makes it easier for molecules to entangle with each other, thereby forming a stable and dense hydration layer that provides pectins with excellent interfacial amphiphilicity and the ability to prevent lipid oxidation. This also explains the superior emulsification ability of RPWP with more RG-I structural domains than that of CP.

### 3.7. Analysis of Emulsion Rheological Properties

To further illustrate the mechanism of RPWPE performance, the viscosity, G′, and G″ were determined, and the results are shown in Figure 6. The viscosity of the emulsions increased with increasing pectin concentrations, exhibiting a dose-dependent effect. According to the results of the rheological property analysis of emulsions, the viscosity of RPWPE was apparently higher than that of CPE; in particular, the emulsion viscosity was significantly higher than that of CPE at 2% concentration when the RPWPE concentration was 1%. Higher viscosity reduces the emulsion fluidity, weakening the free flow of droplets, thus making droplet aggregation less likely and keeping the emulsion system more stable. As expected, the viscosity of RPWPE was greater than that of CPE during the entire process, consistent with the results of the emulsification performance tests. According to Cao et al., pectin molecules with higher molecular weights and complex side-chain structures have more active bonding sites and are more inclined to form net-like structures with high viscosity, which is consistent with the structure of RPWP (Section 3.1) [36].

Because there was no gel network in the CPE at 1% and 0.5% concentrations, G′ and G″ could not be detected. G′ was constantly larger than G″ in RPWPE at 0.5%, 1%, 1.5%, and 2.0% concentrations, and the loss factors (tan δ) are all below 1, indicating the existence of a three-dimensional network structure. The modulus of CP emulsions showed intersections at both 1.5% and 2.0% concentrations, and G′ was surpassed by G″ at high vibrational frequencies, demonstrating that the gel network structure was disrupted with an increase in vibration frequency, by which the emulsion system was transformed from a weak gel to a fluid. In the RPWP emulsion, the G″ of the emulsion gradually approached G′ as the vibration frequency increased but never exceeded it, demonstrating the formation of a more stable gel network in the RPWP emulsion, which enhanced the stability of the emulsion. The higher molecular weight and complex molecular structure of multi-branched chains give RPWP molecules a strong spatial site-blocking effect, creating a spacer layer in the emulsion; acetylation in the pectin molecule makes pectin somewhat hydrophobic, and low DM gives RPWP better interfacial amphiphilicity, making it an effective emulsifier [37,38].

### 3.8. Antioxidant Ability Analysis

Figure 7 illustrates the scavenging capacities of CP and RPWP towards DPPH and hydroxyl radicals, as well as the MDA content in the solution. The hydroxyl and DPPH radical scavenging capacity of RPWP and CP both exhibited a concentration-dependent response within the range 5–20 mg/mL. RPWP exhibited obviously better antioxidant ability than CP, and close to Vc in 20 mg/mL, suggesting that RPWP has the potential as antioxidants to enhance the stability of emulsion through reducing the oxidation of oils in emulsion. According to Zhang et al., high-molecular-weight pectin can form a physical barrier to inhibit relevant oxidatively active enzymes [6]. Moreover, the phenolics in pectin contribute to the absorption of superoxide [39], and because of the characteristics of RPWP, it can provide more protection for the emulsion and exhibit better stability.

As shown in Figure 7, the MDA content in oil was significantly higher than that in CP (1.5%, *m*/*v*) and RPWP (1.5%, *m*/*v*) emulsions on the 15th day, indicating severe lipid oxidation occurred in the corn germ oil group. The growth rate in the MDA content of RPWP (1.5%, *m*/*v*) was slower than that of CP (1.5%, *m*/*v*) over 30 days, suggesting that RPWP exhibited more effective lipid oxidation inhibition than that of CP. The high viscosity of the emulsion and excellent antioxidant effect of RPWP resulted in the decreased oxidation of oils and fats in the emulsion, resulting in better emulsion stability. These results suggest that RPWP has a remarkable antioxidant effect as an emulsifier compared to that of CP, indicating potential for food and medical applications.

## 4. Conclusions

In this study, citric acid and low-temperature water extraction methods were employed to obtain the pectin polymer RPWP, which is rich in RG-I structures and identified as a low-methoxy acetylated pectin. Analytical techniques such as HPGPC, NMR, and methylation analysis revealed that RPWP possesses a higher molecular weight, greater degree of branching, and significant RG-I domain content compared to commercial pectin (CP). These structural attributes enhance RPWP’s usability over CP, as its cross-linking forms a more stable structure, and its branched structure offers additional secondary binding sites for polyphenols and protein substances. Rheological analysis indicated that at a 4% concentration, RPWP exhibits higher viscosity and elastic solid properties. Furthermore, three-phase contact angle testing and micromorphological analysis demonstrated that RPWP has superior interfacial amphiphilicity and a denser surface structure. Emulsions containing 2% RPWP showed good stability over a 30-day storage period and outperformed those containing CP in both ES and EC tests. RPWP also displayed weak gelation and maintained a stable gel network under increased oscillation frequency. In vitro antioxidant activity tests revealed that RPWP, at a 2% (*w*/*v*) concentration, exhibits DPPH and hydroxyl radical scavenging abilities comparable to vitamin C. This strong antioxidant activity enables RPWP to effectively prevent oil oxidation in emulsions, thereby reducing the risk of demulsification. These findings suggest that RPWP can serve as a novel rheology modifier and emulsifier with excellent antioxidant properties; however, further research is warranted to explore its applications in actual products.

## Figures and Tables

**Figure 1 foods-13-02338-f001:**
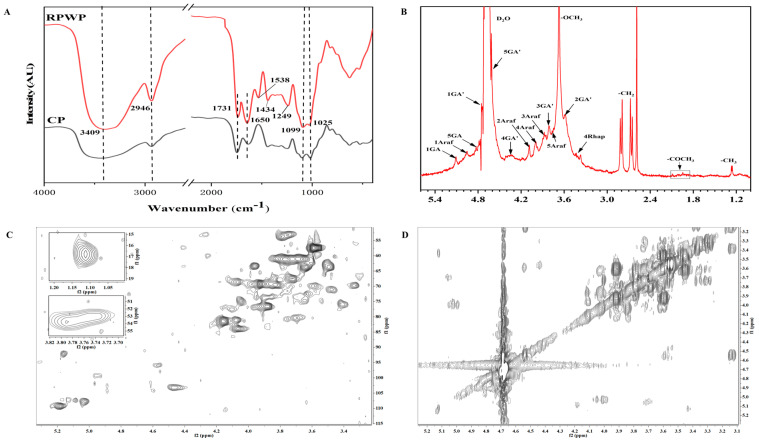
Fourier transform infrared spectroscopy (FTIR) analysis of RG-I-enriched pectic polysaccharides were extracted from watermelon peel (RPWP) and commercial pectin (CP) (**A**), 1H nuclear magnetic resonance (NMR) spectrum (**B**), HSQC NMR spectra (**C**), and COSY spectra (**D**).

**Figure 2 foods-13-02338-f002:**
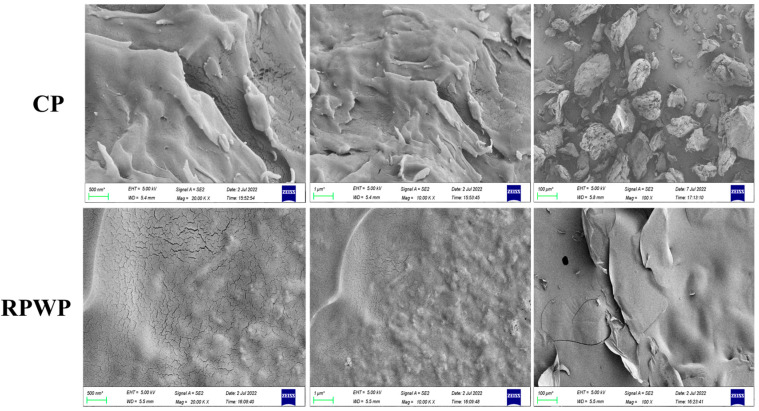
Scanning electron microscopy (SEM) (100×, 10,000× and 20,000×) images of RG-I-enriched pectic polysaccharides were extracted from watermelon peel (RPWP) and commercial pectin (CP).

**Figure 3 foods-13-02338-f003:**
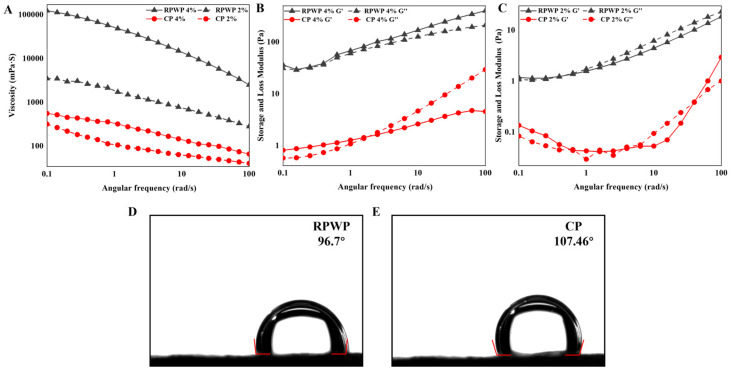
Viscosity curves (**A**) and frequency dependence of storage modulus (G′) and loss modulus (G″) (**B**,**C**) for 2% and 4% pectin solutions and the three-phase contact angles of RG-I-enriched pectic polysaccharides were extracted from watermelon peel (RPWP) and commercial pectin (CP) (**D**,**E**).

**Figure 4 foods-13-02338-f004:**
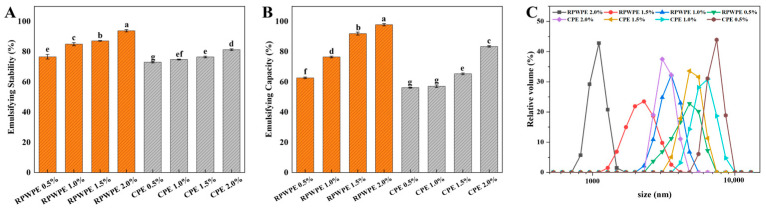
The emulsifying capacity (EC) (**A**), emulsion stability (ES) (**B**), and drop size (**C**) of RG-I-enriched pectic polysaccharides were extracted from watermelon peel emulsion (RPWPE) and commercial pectin emulsion (CPE) (0.5%, 1.0%, 1.5%, and 2.0%). Bars with different letters are significantly different (*p* < 0.05) in the same graph.

**Figure 5 foods-13-02338-f005:**
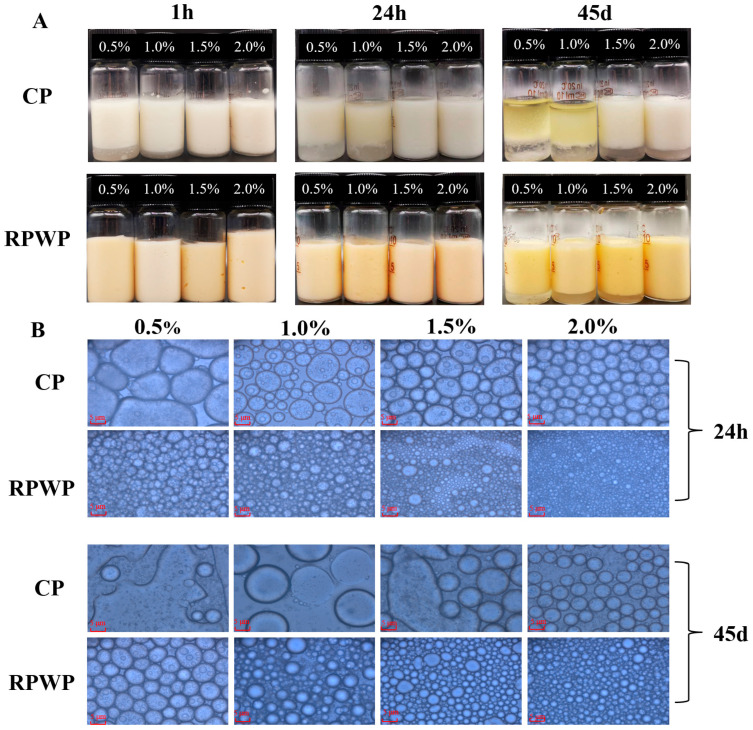
Images of RG-I-enriched pectic polysaccharides were extracted from watermelon peel emulsion (RPWPE) and commercial pectin emulsion (CPE) RG-I-enriched pectic polysaccharides were extracted from watermelon peel emulsion (RPWPE) and commercial pectin emulsion (CPE) oil-in-water emulsions (0.5%, 1.0%, 1.5%, and 2.0%) after storage at 4 °C (**A**) and their optical micrographs (**B**).

**Figure 6 foods-13-02338-f006:**
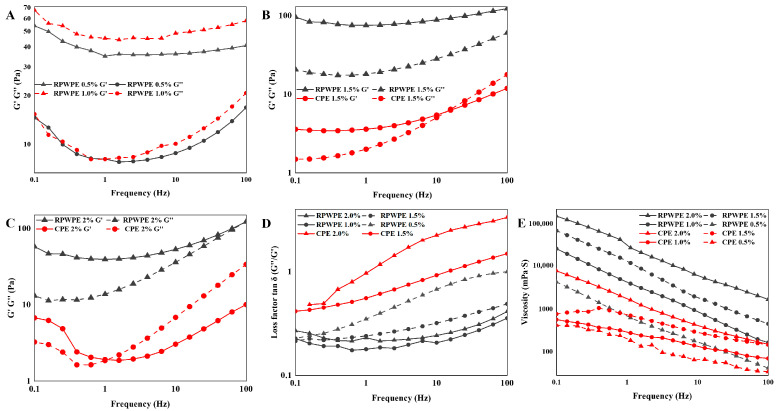
The storage modulus (G′), and loss modulus (G″) (**A**–**C**), loss factor (tan δ) (**D**), and viscosity (**E**) of RG-I-enriched pectic polysaccharides were extracted from watermelon peel emulsion (RPWPE) and commercial pectin emulsion (CPE).

**Figure 7 foods-13-02338-f007:**
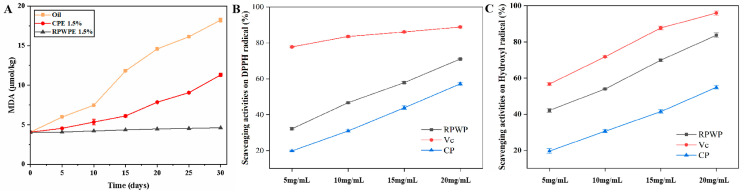
MDA levels in oil, 1.5% RPWPE, and 1.5% CPE during 30 days (**A**). The scavenging activities of RG-I-enriched pectic polysaccharides were extracted from watermelon peel (RPWP) and commercial pectin (CP) solution (2%) for DPPH (**B**) and hydroxyl radicals (**C**).

**Table 1 foods-13-02338-t001:** Monosaccharide compositions and sugar ratios of RG-I-enriched pectic polysaccharides were extracted from watermelon peel (RPWP) and commercial pectin (CP).

Samples	RPWP	CP
**Molecular weight**		
Mw (kDa)	1991 ± 1.1	576 ± 0.45
Mn (kDa)	1025 ± 1.14	324 ± 0.26
Mw/Mn	1.94 ± 0.00	1.78 ± 0.00
**Monosaccharide composition**		
Fuc (mol%)	0.6 ± 0.05	0.3 ± 0.09
Rha (mol %)	2.9 ± 0.12	6.2 ± 0.22
Ara (mol %)	30.7 ± 0.12	2.5 ± 0.26
Gal (mol %)	29.6 ± 0.39	20.1 ± 0.5
Glc (mol %)	3.4 ± 0.09	6.5 ± 1.71
Xyl (mol %)	2.3 ± 0.94	13.2 ± 0.41
Man (mol %)	0.8 ± 0.03	0
GalA (mol %)	27.9 ± 0.51	50.5 ± 0.94
**Sugar molar ratios**		
HG (%)	24.97 ± 0.447	44.27 ± 0.92
RG-I (%)	66.17 ± 0.37	35.00 ± 0.92
Rha/GalA	0.11 ± 0.40	0.12 ± 0.00
(Ara + Gal)/Rha	20.59 ± 0.99	3.65 ± 0.15
GalA/(Gal + Ara + Rha + Xyl)	0.43 ± 0.12	1.20 ± 0.01
Protein (%)	6.1 ± 0.48	2.13 ± 0.28
Total polyphenol (%)	2.4 ± 0.08	2.58 ± 0.07

All results are expressed as “mean ± standard deviation”, *n* = 3.

## Data Availability

The original contributions presented in the study are included in the article/Appendix A, further inquiries can be directed to the corresponding author.

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
