# Peer review of "Pectins Rich in RG-I Extracted from Watermelon Peel: Physicochemical, Structural, Emulsifying, and Antioxidant Properties"

_foods, 2024, doi:10.3390/foods13152338_

Round 1
Reviewer 1 Report
Comments and Suggestions for Authors

Author Response
|
Comments 1: Abstract … RG-I=66.17 %, methylation degree: 41.45 %, (Ara + Gal)/Rha: 20.59 %). |
|
Response 1: Thank you for pointing this out. We agree with this comment. We have corrected the formatting error. (page 1, line 16) |
|
Comments 2: Materials and Methods Line 86, 113, 114, 122: … water bath 60 ºC; for 3 h at 120 °C; … 30 °C; … 0.05 M NaCl solution, Line 157: … at different concentrations (0.5 %, 1.0 %, 1.5 %, 2.0 %, and 4 % w/v) |
|
Response 2: Thank you for pointing this out. We agree with this comment. We have corrected the formatting error. (page 2, line 94), (page 3, line 121, 130), (page 4, line 165) Comments 3: 3 Results and discussions Response 3: Thank you for pointing this out. We agree with this comment. We have corrected the spelling error. (page 4, line 204) |

Reviewer 2 Report
Comments and Suggestions for Authors
In the Introduction, it is suggested to add the objective of the work and not mention all the techniques that were carried out, since they are described in the Methodology section.
The section 3, corresponds to Results and Discussions? or only to Results?
Line 196, Please correct the world:" Reults"
In the title of Table 1, I suggest mentioning what RPWP an CP means.
In the Figure 5: "4oC"refers to 4 °Celsius?
Author Response
|
Comments 1: In the Introduction, it is suggested to add the objective of the work and not mention all the techniques that were carried out, since they are described in the Methodology section. |
|
Response 1: Thank you for pointing this out. We agree with this comment. Therefore, we have revised the introduction part to make it more complete. (page 1, line 34-37), (page 2, line 67-74) |
|
Comments 2: The section 3, corresponds to Results and Discussions? or only to Results? |
|
Response 2: Thank you for your pointing out. In part 3, we combine the results and discussions in one part and we have We have corrected the title of part 3. (page 5, line 204) Comments 3: Line 196, Please correct the world:" Reults" Response 3: Thank you for pointing this out. We agree with this comment. We have corrected the spelling error. (page 5, line 204) Comments 4: In the title of Table 1, I suggest mentioning what RPWP an CP means. Response 4: Thank you for pointing this out. We agree with this comment. We have aadded the meaning of RPWP and CP. (page 6, line 244) Comments 5: In the Figure 5: "4oC"refers to 4 °Celsius? Response 5: Thank you for pointing this out. We agree with this comment. We have corrected the spelling error. (page 12, line 422) |

Reviewer 3 Report
Comments and Suggestions for Authors
The research addresses the possibility of the use of pectins isolated from watermelon peel. The results indicated good antioxidative potential of these pectins allowing wide use in food and pharmaceutical industry. The methodology is well described and the results are presented in an appropriate manner, it just needs some technical improvement by the explanation of the abbreviations in the tables and figures so they would be readable. The conclusions should be more focused on potential and further research in this area than on the repetition of the results. The references are appropriate and suitable for the presented paper.
The paper is well written but needs some minor revision.
Abstract Avoid abbreviations in the abstract
Line 209 please add reference
Line 236 add the abbreviation explanation in the title of the Table
Table 1 add statistiscal analysis to the Table 1
Abbreviations must be explained in all the figures
Conclusion should be rewritten in order to present a summary of the performed analysis and not the repetition of the results
Author Response
|
Comments 1: Abstract Avoid abbreviations in the abstract |
|
Response 1: Thank you for pointing this out. We agree with this comment. Therefore, we have replaced the abbreviations with full names. (page 1, line 13-15) |
|
Comments 2: Line 209 please add reference |
|
Response 2: Thank you for pointing this out. We agree with this comment. Therefore, we have added the reference. (page 5, line 216) Comments 3: Line 236 add the abbreviation explanation in the title of the Table Response 3: Thank you for pointing this out. We agree with this comment. Therefore, we have replaced the abbreviations with full names. (page 6, line 244-245) Comments 4: Table 1 add statistical analysis to the Table 1 Response 4: Thank you for pointing this out. We agree with this comment. We have added the statistical data in 3.1, in this part we analyzed the chemical composition and molecular weight of RPWP and CP. (page 5, line 225) Comments 5: Abbreviations must be explained in all the figures Response 5: Thank you for pointing this out. We agree with this comment. Therefore, we have replaced the abbreviations in all the figures with full names. (page 5, line 244-245), (page 8, line 311-312), (page 9, line 332-333), (page 10, line 361-363), (page 11, line 414-416), (page 12, line 419-421), (page 13, line 454-456), (page 14, line 479-480) Comments 6: Conclusion should be rewritten in order to present a summary of the performed analysis and not the repetition of the results Response 6: Thank you for pointing this out. We agree with this comment. Therefore, we have rewritten the conclusion part to make it better. (page 14, line 482-501) |

Reviewer 4 Report
Comments and Suggestions for Authors
Dear Authors,
The manuscript with title "Pectins rich in RG-I extracted from watermelon peel: Physicochemical, structural, emulsifying and antioxidant properties" is very interesting and deals with extraction of valuable compounds from waste (extraction of RG-I pectin with favorable properties from watermelon rind).
However, I suggest improving of the manuscript in few points:
1. There are English mistakes in the text (for example line 11 Physicochemical should be with small p; reults line 196 should be results). The authors must check spelling mistakes.
2. In the discussion part authors should explain more precisely which part of the branched structure (in terms of monosaccharide composition) is the most responsible for emulsifying properties of Pectins rich in RG-I ?
3. Line 379-380: Why stability of produced emulsions from CPE is unstable? What is difference in the structure of CPE in comparison to RPWPE which is caused significant instability of emulsion after 45 days. Which kind of higher degree of branching is responsible for better rheological properties?
4. Why RPWP showed strong in vitro antioxidant activity, with DPPH and hydroxyl radical scavenging abilities close to vitamin C at 2% (w/v) concentration? Which part of chemical structure of RPWF react more powerful with DPPH reagent in comparison to CP?
I suggest acceptance of the manuscript after major revision.
Comments on the Quality of English Language
There are English mistakes in the text
(for example, line 11 Physicochemical should be with small p; line 196 should be results). The authors must check spelling mistakes and minor editing of English language.
Author Response
|
Response 1: Thank you for pointing this out. We agree with this comment. We have checked the essay and corrected the spelling mistakes. (page 1, line 11), (page 5, line 204) |
|
Comments 2: In the discussion part authors should explain more precisely which part of the branched structure (in terms of monosaccharide composition) is the most responsible for emulsifying properties of Pectins rich in RG-I ? |
|
Response 2: Thank you for pointing this out. We agree with this comment. In Sections 3.2 and 3.3, we analyzed the RG-I branch of RPWP in detail by glycosidic bond analysis and NMR analysis, proving that its RG-I region has a rich branching structure. These rich branching structures provide the RPWP solution with a higher viscosity and are easier to form a gel network structure. These results were confirmed in the rheological analysis of Sections 3.5 and 3.7. In addition, the mild extraction method and rich branching structure enable RPWP to retain a high polyphenol content, giving RPWP excellent antioxidant activity in the emulsion, and the high viscosity allows the pectin molecules to cross-link, providing an excellent encapsulation effect for the oil in the emulsion. These advantages effectively prevent the oxidation of oils and fats, making RPWP have the potential to be used as an emulsifier. Comments 3: Line 379-380: Why stability of produced emulsions from CPE is unstable? What is difference in the structure of CPE in comparison to RPWPE which is caused significant instability of emulsion after 45 days. Which kind of higher degree of branching is responsible for better rheological properties? Response 3: Thank you for pointing this out. Compared with CP, which belongs to HG pectin, RPWP has a richer branching structure in the RG-I region, which makes it easier to cross-link with each other in solution to form a stable O/W emulsion, forming smaller droplets, which can better resist the demulsification caused by gravity. For this issue, we add a more detailed discussion in the analysis to explain it. (page 11, line 389-392) Comments 4: Why RPWP showed strong in vitro antioxidant activity, with DPPH and hydroxyl radical scavenging abilities close to vitamin C at 2% (w/v) concentration? Which part of chemical structure of RPWF react more powerful with DPPH reagent in comparison to CP? Response 4: Thank you for pointing this out. The relatively mild extraction method retains a certain amount of polyphenols in RPWP, providing DPPH and hydroxyl radical scavenging capabilities. In addition, the higher RG-I region content also gives RPWP good antioxidant capacity. This phenomenon has also been found in some previous reports[1-3]. |
|
4. Response to Comments on the Quality of English Language |
|
Point 1: There are English mistakes in the text (for example, line 11 Physicochemical should be with small p; reults line 196 should be results). The authors must check spelling mistakes and minor editing of English language. |
|
Response 1: Thank you for your correction. We have rechecked the spelling errors in the article and corrected them. |
References
- Zhang, S.; Waterhouse, G. I.; Du, Y.; Fu, Q.; Sun, Y.; Wu, P.; Ai, S.; Sun-Waterhouse, D. X., Structural, rheological and emulsifying properties of RG-I enriched pectins from sweet and sour cherry pomaces. Food Hydrocolloid. 2023, 139, 108442. https://doi.org/10.1016/j.foodhyd.2022.108442
- Du, Y.; Zhang, S.; Sun-Waterhouse, D.; Zhou, T.; Xu, F.; Waterhouse, G. I.; Wu, P., Physicochemical, structural and emulsifying properties of RG-I enriched pectin extracted from unfermented or fermented cherry pomace. Food Chem Food Chem. 2023, 405, 134985. https://doi.org/10.1016/j.foodchem.2022.134985
- Zhang, S.; He, Z.; Cheng, Y.; Xu, F.; Cheng, X.; Wu, P., Physicochemical characterization and emulsifying properties evaluation of RG-I enriched pectic polysaccharides from Cerasus humilis. Polym. 2021, 260, 117824. https://doi.org/10.1016/j.carbpol.2021.117824
